# Reanalysis of Trio Whole-Genome Sequencing Data Doubles the Yield in Autism Spectrum Disorder: De Novo Variants Present in Half

**DOI:** 10.3390/ijms25021192

**Published:** 2024-01-18

**Authors:** Omri Bar, Elizabeth Vahey, Mark Mintz, Richard E. Frye, Richard G. Boles

**Affiliations:** 1NeurAbilities Healthcare^®^, Voorhees, NJ 08043, USA; obar@neurabilities.com (O.B.); evahey@neurabilities.com (E.V.); mmintz@neurabilities.com (M.M.); 2Autism Discovery and Treatment Foundation, Phoenix, AZ 85050, USA; drfrye@autismdiscovery.org; 3NeuroNeeds^®^, Old Lyme, CT 06371, USA

**Keywords:** autism, diagnostic yield, DNA sequencing, novel disorders

## Abstract

Autism spectrum disorder (ASD) is a common condition with lifelong implications. The last decade has seen dramatic improvements in DNA sequencing and related bioinformatics and databases. We analyzed the raw DNA sequencing files on the Variantyx^®^ bioinformatics platform for the last 50 ASD patients evaluated with trio whole-genome sequencing (trio-WGS). “Qualified” variants were defined as coding, rare, and evolutionarily conserved. Primary Diagnostic Variants (PDV), additionally, were present in genes directly linked to ASD and matched clinical correlation. A PDV was identified in 34/50 (68%) of cases, including 25 (50%) cases with heterozygous de novo and 10 (20%) with inherited variants. De novo variants in genes directly associated with ASD were far more likely to be Qualifying than non-Qualifying versus a control group of genes (*p* = 0.0002), validating that most are indeed disease related. Sequence reanalysis increased diagnostic yield from 28% to 68%, mostly through inclusion of de novo PDVs in genes not yet reported as ASD associated. Thirty-three subjects (66%) had treatment recommendation(s) based on DNA analyses. Our results demonstrate a high yield of trio-WGS for revealing molecular diagnoses in ASD, which is greatly enhanced by reanalyzing DNA sequencing files. In contrast to previous reports, de novo variants dominate the findings, mostly representing novel conditions. This has implications to the cause and rising prevalence of autism.

## 1. Introduction

Autism spectrum disorder (ASD) is a complex developmental disorder with early onset, manifesting as deficits in social communication and interaction and the presence of restrictive/repetitive behaviors and interests and/or sensory disorders that interfere with daily functioning. Although ASD is a neurobiological disorder of early brain development, diagnosis is currently based on a behavioral phenotype with no convincing evidence of consistent biomarkers, suggesting biological heterogeneity [1]. Multiple studies have demonstrated strong heritability components in ASD, suggesting underling genetic mechanisms (biological vulnerability, [2,3,4]). However, substantial asymmetry in the phenotype of first-degree relatives carrying the same major-disease-associated DNA variant, and frequent acute or subacute development of ASD features following a physiological stressor, suggest the addition of strong environmental components (triggers [5]).

In many cases, close relatives of people with ASD are themselves affected with a neurodevelopmental disorder or a forme fruste phenotype. Inherited highly penetrant variants in the proband and relative(s) can be identified using DNA sequencing that segregates into a Mendelian (e.g., autosomal recessive or dominant, X-linked) or non-Mendelian (e.g., polygenic, maternal) inheritance pattern. In many other cases, the family history is unremarkable and other mechanisms are identified or assumed (e.g., de novo variants, multiple low-penetrance genes, or a predominately environmental etiology). 

The key role of rare, highly penetrant variants, either inherited or de novo, in the development of ASD has been established by many studies, leading to many proposed mechanisms and associated genes [6,7,8,9,10,11,12,13,14,15,16]. For instance, whole-exome sequencing (WES), which identifies genetic variants in the ~1% of the genome that encodes proteins, found 50 genes as potentially associated with ASD through analysis of de novo protein truncating variants that only occurred in probands [17]. ASD-specific online references, such as SFARI [18] and AutDB [19] have accrued hundreds of genes associated with ASD, and the gene lists are rapidly expanding. 

The clinical use of genetic testing in ASD has recently been reviewed [20]. The diagnostic yield of studies varies widely as larger proportions of the genome are evaluated. While Fragile X syndrome appears to have a stable diagnostic rate (yield) at about 1–2% [21,22], the yield of copy number variants has demonstrated substantial variation among studies, ranging from 3 to 41% [22,23,24,25]. In the early days of next-genome sequencing, targeted approaches were common; one such study in 208 candidate genes revealed a yield of 6^ [26]. WES queries for single-nucleotide (nt) and other small (<about 50 nt) variants among all protein-coding genes; yield was 9 and 16% in two studies [27,28]. One outlier study with a yield at 90% did not consider conservation or clinical correlation and had a high proportion of autosomal recessive findings, consistent with the elevated rate of consanguinity in the Saudi population [29]. However, these types of genetic studies have their limitations, most significantly including lack of coverage of the whole genome, including the mitochondrial DNA (mtDNA), and only considering single-gene etiologies. 

The last decade has seen dramatic improvements in the utility of whole-genome sequencing (WGS) due to advancements in methodology, bioinformatic tools, and variant databases. WGS holds the potential to extend rare-variant discovery to the ~99% of the genome that is noncoding, leading to the potential use of genetics in personalized diagnostics and treatment. Three WGS studies in ASD had yields between 33 and 50% [30,31,32].

Essentially all such studies to date in ASD have been conducted in highly-specialized university centers. The yield of genetic testing in independent healthcare settings using commercial laboratories has not been assessed. By design, the standardized bioinformatic pipeline in commercial laboratories does not report variants in non-established autism genes, and we hypothesized that an in-depth review of such variants would improve diagnostic yield. Thus, in this study, we explore the current yield of trio (patient + biological parents)-WGS in 50 unrelated subjects with ASD, followed by a raw-data analysis, in an independent healthcare organization. We demonstrate high yields of trio-WGS, especially for de novo variants in genes not previously reported in ASD, and not included in a standard laboratory report. 

## 2. Results

### 2.1. Subject Characteristics

Among our 50 unrelated subjects, 12 (24%) were female. The age at the time of sequence review ranged from 4 to 26 years, with a median of 12 years. Mean maternal and paternal ages at the subject’s birth were 33 and 35 years, respectively. Thirty-five (70%) were of Western Eurasian ancestry (22 European Americans, 2 Ashkenazi, 3 European/Ashkenazi, 1 Armenian, and 7 “White” not otherwise specified). Eleven (22%) were of other backgrounds, including 5 South Asians, 1 East Asian, 2 African Americans, and 4 of mixed-race from Latin America. Two subjects were of mixed Western Eurasian and other (Central Asian, African) ancestry. Finally, the race of 2 subjects was not recorded. Given the predominance of Western Eurasian ancestry and the large degree of ancestral heterogeneity in the remainder of our patient population, we used gnomAD data for Non-Finnish Europeans (NFE) as the comparison group. NDD diagnoses beyond ASD and non-NDD diagnoses were common among our subjects, as shown in Table 1.

### 2.2. De Novo Variants

As defined in Methods, 46 de novo coding variants were identified in our 50 subjects, including 31 in genes directly related to ASD, 13 in genes indirectly associated with ASD, and 2 identified in genes without known ASD association (Appendix A). To decrease false-positive results, we decided to only consider as Primary Diagnostic Variants (PDVs) those variants in genes with direct association with ASD (Direct genes). PDVs as we defined are highly likely to be disease related in our subjects. The algorithms for determination of PDVs and Direct genes are detailed in the Subjects and Methods section. Among the Direct genes, 29/31 de novo coding variants (94%) are Qualifying variants, versus 5/15 (33%) of the variants in genes without a direct association (non-Direct genes − indirect + non-associated) (*p* = 0.0002, odds ratio 29, 95% C.I. 4.8–170).

Qualifying de novo coding variants that met our criteria for PDVs were identified in 25 (50%) of the 50 ASD probands (Table 2, column 4). Six variants qualified in the A1 category, with missense variants in two genes (EHMT1, KCNB1), a deletion including an A1 gene (UBE3A in subject #9), an in-frame amino acid deletion (SPEN), one frameshift (ANKRD11), and one truncating (ANKFN1). Six variants qualified for the A2 category, including three missense (GRB10, GABRA1, AGO3), two deletions including A2 genes (OTUD7A, FAN1, TRPM1, ARHGAP11B, CHRNA7 in #7; GGNBP2 in #32), and one frameshift (RIF1). Sixteen variants qualified for the A3 category, with thirteen missense (COL4A1, TRPM2, MTMR4, USP20, GRIK1, CEP170, NUP210, KRAS, YTHDF1, STAT1, GOLGB1, SP8, HSPA1A), one truncating (PGAM5), one splice acceptor (SLC41A2), and one frameshift (OCM). One patient had two de novo variants that qualified as PDVs.

Neither maternal (with 33.2 ± 4.9 yrs, without 33.8 ± 4.0 yrs, *p* = 0.7) nor paternal (with 34.4 ± 7.1 yrs, without 36.2 ± 4.7 yrs, *p* = 0.3) age varied based on the presence or absence of a de novo PDV.

### 2.3. Inherited Variants

As there are tens of thousands of inherited variants in the exome and nearby areas alone in all individuals, whether affected with ASD or otherwise, we only tabulated those that meet our criteria for Qualifying variants in genes directly related to ASD. These variants are shown in Table 2, column 5, with italics labeling those that met our criteria for PDVs. Ten of the probands (20%) carry Qualifying inherited variants that met our criteria for PDVs. This includes four probands with autosomal recessive variants, including two homozygous (SLC1A4, EIF3F) and two in trans compound heterozygous (IVD, ZNF292). X-linked variants (THOC2, USP9X, NLGN4X), whereas the male proband is hemizygous and the mother heterozygous, are present in three subjects. Autosomal dominant variants with a parent affected with significant neurodevelopmental disorder were identified in two families (TCF20, paternally inherited, and RERE, maternally inherited). Lastly, one subject was found to have a mitochondrial variant (MT-CO1 m.6082C>T, 58% heteroplasmy) that qualified as a PDV due to heteroplasmy and a significant matrilineal history (maternal ratio 4.0, maternal inheritance ratio 6.67, [33]). The latter subject also had a de novo PDV. All coding inherited Qualifying variants identified among our subjects are shown in Appendix A, depending on their mode of inheritance, with copy number variants in Appendix A.

### 2.4. Combined Primary Diagnostic Variants and Yield from Laboratory Report

Combining both de novo and inherited variants, at least one PDV was identified in 34 subjects (68%, Table 2). PDVs in genes directly associated with ASD (Direct genes) were far more likely to be Qualified than non-Qualified, versus our control group of non-Direct genes with lesser association (*p* < 0.0001, odds ratio 43, 95% C.I. 4.4–420). All variants considered are listed in the Appendix A, including the basis of annotation as Qualifying or non-Qualifying, as well as the data we used to score the genes in regard to their published association with autism in categories ranging from “A1” to “B3”.

Consistent with our hypothesis, only 14/34 (41%) cases with at least one PDV had the variant listed on the laboratory report (Table 3), including 9/25 (36%) de novo and 6/10 (60%) inherited PDVs, for an overall yield of 28% (14/50).

### 2.5. Genotype–Phenotype Correlation

A PDV was identified significantly more often in subjects with neither tics nor OCD (22/26 (85%) versus the presence of either 12/24 (50%), *p* = 0.01, odds ratio 5.5, 95% CI 1.5–21). Neither manifestation alone was statistically significant; although, there was a negative trend for PDV identification in subjects with tics (*p* = 0.08). All other clinical comparisons were non-significant, including no relationship with sex, the severity of ID, or the presence of seizures. A trend was observed for PDVs being less common in subjects with absent speech (6/13 (46%) vs. 28/37 (76%), *p* = 0.08). In another potential trend, a PDV may be less common in patients with developmental regression (with 16/27 (59%) vs. without 18/23 (78%), *p* = 0.2). Cases with and without a PDV are not significantly more or less likely to have an affected first-degree relative with a substantial NDD; although, there is a possible trend (+PDV 13/34 (38%) with a first-degree relative affected, no PDV 3/16 (19%), *p* = 0.18, odds ratio 2.7, 95% CI 0.6–11). Since there are few Mendelian cases, comparison of cases with and without a de novo PDV are essentially identical to the above. In particular, there is no difference regarding those with mild versus severe intellectual disability compared to those with or without a de novo variant (*p* = 1.0000). However, such correlations are difficult given the relatively small number of subjects in this study.

### 2.6. Candidate Polygenic Modifier Variants

Inherited, Qualifying nuclear variants in Direct genes directly associated with ASD that did not meet our criteria for PDVs were defined as Candidate Polygenic Modifier (CPM) variants (genes in Table 2, column 5, non-italic font; variants in Appendix A). Twenty five of 34 subjects with a PDV (73.5%) were found to also have at least one CPM, versus 11 of 16 subjects without a PDV (69%, P = NS). Viewing the same issue with the number of variants, we noted equal frequencies in cases with and without a PDV (1.3 Qualifying variants per subject in both groups).

### 2.7. Actionability of Genetic Results

In 3 subjects, no variants of interest (PDV or CPM) were identified, and in an additional 10 cases, at least one variant of interest was identified, but no management changes were made based on that information (non-actionable). In the remaining 37 subjects (74%), DNA data were directly actionable, including 33 cases (66%) in which either a non-prescription supplement and/or a prescription medication were recommended based on genetic results. Supplements were recommended based on DNA findings in 30 subjects (60%), with 3 or more cases consisting of combination mitochondrial-targeted products, individual antioxidants, omega-3 fatty acids, potassium, magnesium, zinc, and 5-hydroxytryptophan. Medications were recommended based on DNA findings in 24 cases (48%), with 3 or more cases consisting of memantine (and/or other NMDA antagonists), acetazolamide, propranolol, and verapamil (and/or other calcium channel blockers). Multiple other therapies were limited to one or two subjects each. Clinical-outcome data are being collected. Additional laboratory testing was ordered in six cases (12%) based on the results of “on-target” (likely related to autism) genetic testing results. In addition, nine subjects were referred to a specialist based on DNA results, including eight to Immunology. 

Thirteen subjects (26%) were identified with variants of definite or probable clinical significance that were assessed by the corresponding author to be incidental (“off target”, likely not related to the patients’ ASD). Among these, four had Pathogenic variants in the FLG gene, which is associated with a wide spectrum of skin disorders ranging from eczema to ichthyosis. Four (8%) were identified with variants in cancer predisposition genes, including two with APC (mild increased risk), and one each with BARD1 and MUTYH; the latter of which was previously identified in the family. Two subjects were identified with variants of uncertain significance in the VWF (Von Willebrand) gene, and the appropriate blood testing was recommended. The remaining off-target findings were a Pathogenic HFE variant where a grandparent has hemochromatosis, a Pathogenic GJB2 variant in an individual with severe hearing loss, mosaic monosomy X (Turner syndrome) which was previously identified via other testing, and a variant in PER3 in an adolescent with a significant sleep disorder. All off-target variants identified among our subjects are shown in Appendix A.

## 3. Discussion

### 3.1. Our Subjects Represent the Broad Phenotype of Autism in Terms of Sex, Severity, and Co-Morbidities

The clinical data (Table 1) illustrate that our 50 subjects are a good representation of the spectrum of ASD often presenting for medical evaluation. The sex ratio of 22% female is in accordance with ASD being diagnosed approximately four times as often in males [34]. One frequent criticism of Neurology- or academic-ascertained patient-derived populations is a skew towards more severe and complicated cases. Like most disorders, the ASD spectrum can be visualized as an iceberg, with fewer severe cases, more moderate cases, and a hard-to-identify mild majority “under the surface”. Regarding the latter, many are forme frustes that do not meet clinically determined diagnostic criteria. 

While ASD is a complicated syndrome, severity categorization of “high” versus “low” functioning is usually determined by the presence/absence and degree of intellectual disability (ID). In Table 1, column 3, the degree of ID among our subjects is shown with highlighting reflecting cases of moderate to greater levels of ID, while the absence of highlighting reflects cases with lesser degrees through normal cognition. The line separating the more “severe” from more “mild” cases was drawn to best separate individuals who generally require constant ongoing supervision, versus those that can have at least some degree of independent living. With this boundary, exactly half of our cases (25) are severe, while the remaining half are mild. 

Another parameter that clinically can be used to separate severe from mild cases is the presence or absence of functional speech. Many patients with ASD have absent to near-absent speech such that verbal communication is nearly impossible. Thirteen cases (26%) are so highlighted (Table 1, column 4), which is within the 25–30% range often quoted [35,36]. Fifteen cases (30%) are highlighted (column 5, Table 1) with seizure disorders, while another two borderline cases have lesser highlighting (total 34%). These figures align closely with the one-third proportion often quoted with epilepsy [37]. Developmental regression in ASD is common and pleomorphic [38], which is reflected in our population (column 6) as 11 cases (22%) with multiple or greater episodes of regression and 16 cases (32%) with lesser episodes (54% in total). Also, as is common in autism, gastrointestinal (GI) manifestations (42%), OCD (32%), anxiety (22%), and ADD/ADHD (18%) are frequent in our subjects, all of which may be underestimates as these issues were neither systematically queried nor tested for. 

### 3.2. WGS with Comprehensive Sequence Reanalysis Revealed High Sensitivity for Identification of Primary Diagnostic Variants (PDVs) in Our Autism Subjects

We defined Primary Diagnostic Variants (PDVs) conservatively to define genetic variants that are highly likely to have a strong association with ASD in that individual. Variants assigned as PDVs all were “Qualifying”, defined as affecting the amino acid sequence (“coding”), being very rare in humans, and highly conserved at least through mammalian evolution, with the latter rarely being required in other genomic studies. We chose a threshold of mammalian evolution as we believe that variants leading to ASD should not be tolerated among other mammals with similar brain architectures, but might be tolerated beyond given more-pronounced differences in other vertebrate classes. Additionally, the gene must have a published direct association with ASD (e.g., differentially expressed in ASD brain, as discussed in Methods), meets clinical correlation, and the subject’s pedigree is consistent with the mode of inheritance of that variant and gene, if known. Variants that meet all of the above requirements can be considered as highly likely to be disease related, corresponding to a major genetic predisposition in each individual, without which autism is unlikely to have occurred.

Our finding of a de novo PDV variant in 25 subjects, fully in 50% of our cases, is substantially higher than that previously reported in studies on ASD (Table 4, combining the prior three WGS reports [30,31,32], *p* < 0.0001 compared to our data). 

Coding de novo variants (DNVs) are rare in control populations, although they do occur at a low rate, and are not all disease related (e.g., PDVs). As we found that DNVs in Direct genes directly associated with ASD were far more likely (odds ratio 29, *p* = 0.0002) to be Qualified than non-Qualified, versus non-Direct genes with lesser association, our data suggest that the vast majority of our de novo PDVs (all of which are Qualified variants in Direct genes) are disease related. Indeed, we likely were too strict and excluded disease-associated DNVs that did not meet our PDV criteria. Among the 46 de novo coding variants identified in our 50 subjects, 44 (96%) are in ASD-associated genes, including 13 in genes indirectly associated with ASD. While the number of genes with known ASD association is difficult to tabulate; yet certainly large, the proportion is highly unlikely to approach 96% of the 23K+ number of protein-coding genes. Thus, de novo coding variants in our ASD population are highly enriched in ASD-associated genes. 

Among our 50 subjects, 12 DNVs (0.24 per individual) were detected that are unlikely to be disease associated based on being non-Qualifying (11) or Qualifying in a “B2” or “B3” gene without a published association with ASD (1). Given an observed DNV rate of 1 × 10^−8^ per base pair [39] and the exome size of 3 × 10^7^ bp, which results in an expected exome DNV rate of 0.3 per person. Further refining to remove silent variants provides an estimate of 0.2 DNV/individual, near equivalent to the findings in our subjects. 

Ten cases (20%) had inherited PDVs corresponding to known conditions. Our subjects assigned with inherited PDVs all demonstrated good clinical correlation with published cases, a requirement difficult to achieve in very large studies or in the absence of full clinical records. Family histories also were consistent with the known mode of inheritance. While that requirement could and often was met by sporadic disease in the proband for cases with recessive disease, disease manifestation in a relative(s) was required for dominant and maternal inheritance patterns. Overall, our finding that 20% of ASD cases can be assigned to a known inherited condition is in line with previous reports in ASD [27,29,30,31,32]. In particular, the three WGS studies [30,31,32] assigned causality to inherited variants in 32%, 27%, and 31% of their subjects, combining the data that is 21% (50/233 subjects), a figure virtually identical to our data. 

Adding the inherited and non-inherited (de novo) variants together, a PDV was identified in 34 of our present subjects (68%). The authors stress that this yield was obtained following our Comprehensive Sequence Reanalysis of raw data from the laboratory, which increased the yield of identifying a PDV from 28% to 68% of the subjects. If only variants listed on laboratory reports as “Pathogenic” or “Likely Pathogenic” are considered, only 7/50 (14%) of ASD cases were diagnostic. Among the 20 PDVs “missed” by the laboratory, 16 are DNVs. The vast majority are in “Genes of Uncertain Significance” as defined by the clinical laboratory, including 14 cases in genes with no prior reports of disease-associated variants (novel etiologies, Table 3). While 11 of those genes are listed by HGMD with 1–5 neurodevelopmental-disease cases (Table 3); these are either unreported or reported in large studies with little to no variant information, and thus may not be disease related. In addition, no clinical information is available, and thus the present report comprises the first true cases of ASD reported as associated with these 14 genes. An additional three cases were identified in genes for which five or fewer cases were published in association with ASD (very rare etiologies). The identification of both categories is outside of the purview of Variantyx, or any, clinical laboratory. The vast majority of known/established conditions identified using our intensive reanalysis were written on the laboratory report (15/19, 79%, Table 3).

A PDV was identified significantly less often in subjects with either tics or OCD, and a potential negative trend was noted regarding developmental regression. Since these manifestations are all cardinal features of PANS/PANDAS, the data may be suggesting that cases with manifestations associated with these post-infectious immunological entities might be less likely to be primarily genetic in etiology, or we do not know what genetic variants to look for.

In four of the subjects (#10, 20, 29, and 44; 8% of the total), we did not find any variants that are likely correlated to autism using our methodology. There are many potential explanations for this, including genes not yet associated with ASD, variants dismissed for one of various reasons as described in the Methods that indeed are disease related, variants not well elucidated or evaluated by WGS using current practices (e.g., repeat areas, non-coding), multiple common variants, and/or epigenetic/environmental factors.

### 3.3. Autism as a Polygenic/Multifactorial Condition

In the 16 subjects without a PDV, at least one CPM variant was identified in 11; with two or more CPMs in 9 of those. It is tempting to assign these cases as examples of “polygenic” inheritance in ASD, versus the PDV cases which are “monogenic”. In that model, these polygenic cases would be expected to have more CPMs, and higher proportions of affected relatives, versus the monogenic cases (which are mostly de novo or recessive). However, CPMs were identified in a near-equal proportion of subjects, as well as the number of CPMs per individual, in cases with and without a PDV. Furthermore, cases without a PDV are no more likely to have an affected first-degree relative; indeed, there is a possible trend in the other direction. 

Instead, we propose that nearly all of our cases are polygenic in terms of genetic predisposition, with the PDVs likely constituting a high proportion of the genetic component in disease pathogenesis among those cases. In this model, the CPMs function as per their name, as candidate polygenic modifiers of disease, with the primary factor in disease being the PDV, environment, and/or absent (highly multifactorial cases). Our finding of two PDVs in two subjects also is supportive of polygenic inheritance. High frequency/low prevalence variants (“genetic background”) may also have a modifying role but would require a greater number of subjects to evaluate than are available to us, and thus were not analyzed in this study. 

It is no surprise to active clinicians in the field that ASD is not generally, or perhaps ever, a monogenic disease. Indeed, not a single variant has been well characterized that can cause ASD in and of itself. Even well-established ASD-related genes and variants are not fully causal/monogenic conditions in terms of ASD, including the trinucleotide repeat in FRM1 causing Fragile X syndrome, chromosomal deletions across the maternal UBE3A gene causing Angelman syndrome, and loss-of-function variants in SHANK3 causing Phelan-McDermid syndrome. In each case, a substantial proportion of affected individuals have ASD, yet a substantial proportion do not, with observed discordance even among siblings. Also, in these cases a substantial proportion of affected individuals have syndromic manifestations (e.g., birth defects, dysmorphia, small size), and yet a substantial proportion do not, blurring the line between “syndromic” and “primary”/“non-syndromic” cases. Discordance between and within families is presumably polygenic and/or environmental in origin.

Three of our subjects have a sibling affected with a significant neurodevelopmental disorder (as defined in Section 4) that also underwent WGS and extensive raw data analysis by the corresponding author. Each of those families is “asymmetrical”, meaning that one sibling (the study subject) is far more clinically affected, including in terms of ID. In the first family, a de novo copy number PDV was identified in the greater-affected sibling only (1.6 Mb deletion including GGNBP2), and two CPMs were identified in both affected brothers (in DEAF1 in an “A1” gene and SETD1B in an “A2” gene). In the second family, the greater-affected sister has a de novo PDV in SPEN (“A1”), while the lesser-affected sister has a de novo PDV in DNAH14 (“B1”), and both have significant heteroplasmy (58% each) in the mtDNA gene CO1 (a Direct gene, but difficult to subclassify mtDNA). In the third family, the male subject is hemizygous for an X-linked PDV in THOC2, and his mildly-affected sister is heterozygous. In addition, both are compound heterozygous for variants in RIC1 (“A3”). While the numbers are small, certainly phenotypic asymmetry among ASD siblings can have a complex genetic basis. This is consistent with and can explain the finding that 69% of siblings with ASD and identified genetic disorders have different genetic mutations [40]. Indeed, previous studies like this assume a monogenetic cause and fail to appreciate the modifying variants that might be common to both siblings. This has counseling implications for autism-recurrence risks in siblings, even if a DNV is identified.

The fact that the 36 PDVs we identified are located in 36 different genes underscores the emerging reality that sequence variation in a great number of genes can predispose towards ASD. However, the main known function(s) of the majority of those 36 genes cluster in a small number of specific pathways (Table 3). The seven pathways illustrated in Table 3 are all well established as associated with ASD, and include cation transport, redox state (including mitochondrial energy metabolism), amino acids (metabolism and transport), ubiquitin (a major protein-degradation pathway), neurotransmission, gene expression (both general and neuronal targeted), and cell division. What these pathways have in common is their fundamental importance to life, with the first five being preferentially critical to neurons. 

Another fundamental aspect of ASD known to active clinicians in the field is that disease severity in many cases is contemporaneously associated with physiological stressors, especially infections. Other potential stressors that have been observed include medications, toxins, and vaccinations. Common responses are the acute or subacute onset of developmental regression, epilepsy, movement disorders (including tics), and/or psychiatric disease (especially OCD), as well as exacerbation of cardinal traits associated with ASD, closely following the stressor. While the most extreme cases often receive a diagnosis of pediatric acute-onset neuropsychiatric syndrome (PANS), there is no clear boundary between this entity and ASD, and acute/subacute, anecdotally triggered, disease progression is quite common in ASD. Thus, environmental factors, epigenetic changes, and gene–environmental interactions must also be considered in the putative multifactorial pathogenesis of ASD. Indeed, a proportion of our cases without a PDV might be substantially environmental in terms of pathogenesis, which is reflected in the trend towards fewer affected first-degree relatives. None of the findings in this study exclude an important environmental factor in the pathogenesis or pathophysiology of ASD. Genetic changes are present at conception, or very soon thereafter; yet, genetic expression is heavily modified by the environment. Our findings (PDVs and CPMs) are presented as the genetic factors predisposing some individuals towards the development of ASD, under the influence of the rest of the genome and the environment. Environmental factors are also a primary hypothesis for the dramatic increase in ASD prevalence in the past few decades. While our genes change only very slowly, our environment recently has undergone dramatic alterations in many respects.

### 3.4. Variant Curation Comparison to ACMG Guidelines

We chose to develop and use a novel system of variant curation, instead of using the standard American College of Medical Genetics and Genomics’ (ACMGGs’) guidelines [41]. Those guidelines are in general use by laboratories for variant curation in clinical cases, whereas that gene is known to be associated with a condition present in the patient. The main finding of our study is the high prevalence of de novo variants in ASD, most of which are in genes not known to be associated with that condition. In order to reduce false positives induced using the ACMGG guidelines across all 23,000-plus genes, we added the additional requirements of very low prevalence, conservation through mammals, and a published direct link of the gene/protein to ASD. All 27 de novo PDVs we report (Table 3) at least meet ACMGG criteria for Likely Pathogenic (based on PS2/DNV and PM2/not present in control individuals), with 8 of those variants meeting criteria for Pathogenic (adding PVS1/null variant). The only caveat is that we apply PM2 for prevalence < 0.00001 (one in 10 K alleles). ACMGG guidelines were published in 2015 when control sequences were limited, while current databases constitute hundreds of thousands of such individuals. Since mildly-affected or non-penetrant individuals cannot be excluded from these databases, and even severely-affected individuals may have been inappropriately included, using absolute zero prevalence in order to apply PM2 is no longer appropriate, in our opinion. Eight of twenty-seven de novo PDVs had allele prevalence < 1/10,000, but not zero. Using an intermediate cutoff for PM2 of less than one in 100 K alleles, the DNVs in subjects #4 and 26 would have been curated as Variants of Uncertain Significance.

Applying ACMGG to the 12 inherited PDVs in 10 subjects (Table 2 and Table 3), results in 2 Pathogenic (in subjects #2 and 13) and 10 Variants of Uncertain Significance (VUSs). Standard Clinical Genetics practice evaluates VUSs for clinical correlation, and seriously considers those for potential diagnoses that are highly plausible based on the characteristics of the variant (e.g., prevalence), patient (clinical correlation), and family history. This is exactly what the present authors did in assigning variants as PDVs. All 10 genes had adequate clinical descriptions in the literature to allow for robust clinical correlation.

### 3.5. Limitations of the Study

One limitation is the sample size of 50 patients. Rapid developments in science and technology complicate any multi-year study into the effects of genetic testing, as it would be obsolete before publication. Our subjects represent all patients meeting inclusion criteria over 15 months of clinical practice. Furthermore, larger studies ascertained from multiple clinicians or databases often lack detailed and consistent clinical information and methodologies. 

Despite our diagnostic yields for DNVs being substantially higher than that reported in other studies, there are likely to be missed causal variants due to an incomplete sensitivity of our methodology. Four Qualifying variants in B1 genes (with published indirect association with autism) were identified in four subjects, in the following genes: BRPF3, GTF2A1, POGLUT3, and TMEM184B (Appendix A), including a de novo loss-of-function variant in POGLUT3. Each of these variants would have been annotated as PDVs if there was a single study directly linking it to ASD. Certainly, some of these are likely PDVs, conferring substantial predisposition for these individuals to develop autism. Others may act more as CPMs. 

Further affecting sensitivity, additional genetic mechanisms might apply to ASD that were not included herein. For example, a loss-of-heterozygosity (LOH) variant was identified in one of our subjects (#11) with 12 million bp LOH across the centromere at 11p11.2q12.1 [46,885,688-52,819,559], which could be due to uniparental disomy or unknown distant consanguinity. Non-coding variants, particularly DNVs, were not considered in this paper despite being implemented in autism [42] as they are far more, and require a much greater number of study subjects for analyses. As explained, low penetrance/high prevalence variants were not included for the same reason. Finally, environmental causes were not assessed in this study, although eight cases (16%) had a clinical course similar to PANS and either had CPMs in immunological-related genes, or essentially negative genetic findings. These latter cases likely have primary environmental etiologies (infection) and were referred to Immunology for further work-up and treatment. 

Based on our data, 14 additional genes can be added as likely associated with ASD, all based on Qualifying de novo variants. While our statistical analysis suggests that the vast majority are likely disease causal, the lack of functional assays is a limitation. Since all of these genes had prior direct association with ASD, these data do not expand knowledge on potential pathways as the previous work was based on prior data and assumptions. However, new connections might be made regarding our findings regarding de novo variants in 12 of the B1 genes (Appendix A).

### 3.6. Risks and Additional Costs

The families’ insurance company paid for molecular testing in almost all cases as trio-WGS/WES in ASD is generally covered in the USA. In some cases, the insurance company paid for WES, and Variantyx provided WGS. 

Families, physicians, and payers are sometimes reluctant to order WGS out of fear of identifying many potential problems that result in additional testing, referrals, and diagnoses, resulting in higher costs and anxiety. However, our results demonstrate that additional testing, referrals, and off-target diagnoses stemming from the extensive genetic data were few and only in a minority of subjects. Indeed, substantial additional testing was only ordered/recommended based on WGS in a single case, whereas a DNV in COL4A was identified indicating potential “malignant” vascular Ehlers-Danlos. Actionability included avoidance of contact sports. Off-target information with potentially serious clinical implications that was previously unknown to the family was provided only to one family, with the finding of an inherited variant in BARD1 indicating a highly increased cancer risk. As a result, management plans for disease surveillance were instituted/recommended that may mitigate those potential risks. 

### 3.7. Implications of Our Data to a Greater Understanding of ASD

How can our finding of DNVs as the predominant genetic factor in ASD be reconciled with the rapidly-accelerating incidence of this condition? While the modern trend of having children later in life, particularly advanced paternal age, might be a minor factor therein, this was not observed in our relatively small sample size. A much-larger recent study [43] has suggested that advanced paternal age increases DNV rate, but only mildly (48 DNVs per genome at age 25 versus 65 DNVs at age 42), which is far too insufficient of an effect to account for the exploding prevalence of autism. Genetic changes affecting the DNV rate (e.g., encoding DNA repair proteins) also cannot account for this as a species’ genome changes only slowly over time. More likely, DNV rates are accelerating due to some component in our rapidly, and profoundly, changing environment. One potential hypothesis includes environmental toxins, of which there are many potential candidates. One candidate is cadmium, which can induce DNA damage via mitochondrial dysfunction and interacts “with DNA repair mechanisms, cell cycle checkpoints and apoptosis as well as with epigenetic mechanisms of gene expression control” [44]. Another possibility, folate, appears to increase the DNV rate both in deficiency and in substantial fortification [45]. 

An alternative hypothesis is that the DNV rate is relatively static over time, and these variants affect brain homeostasis, but disease may or may not develop dependent on the environment. For example, one of the common pathways frequently seen with DNVs in ASD is the redox state/mitochondrial function, which can be affected by multiple environmental factors, including “polycyclic aromatic hydrocarbons, air pollutants, heavy metals, endocrine-disrupting compounds, pesticides, and nanomaterials” [46]. Additional studies are warranted to replicate our finding of a high rate of DNVs in autism, and to determine whether the population’s DNV rate is indeed increasing, and if so, why?

## 4. Subjects and Methods

### 4.1. Subjects

A review of all patient notes written by the corresponding author was conducted to determine study eligibility. Our subjects consist of the 50 most-recently evaluated, sequential, unrelated patients with a clinical diagnosis of ASD in which trio-WGS was performed at Variantyx^®^ (Framingham, MA, USA). Each subject was evaluated by the corresponding author, who is a clinical geneticist and pediatric quaternary care specialist known for conducting clinical care and research. The minimum requirements of this evaluation included a chart review, interview of the parent(s) (and of the patient as feasible), and a physical examination (via video-teleconference) The diagnosis of ASD in each case was confirmed by another specialty provider (e.g., child neurologist, developmental pediatrician, psychologist) as well as by appropriate neuropsychological or neuropsychiatric testing. Subjects with additional neurodevelopmental diseases (NDD) or non-NDD diagnoses were not excluded, which was present in all cases. In the few cases where more than one family member met study criteria, the subject was assigned to be the proband (person first presenting as a patient). In cases of affected siblings presenting simultaneously, the elder was assigned. Thus, all study subjects have no known genetic relationships to each other.

This study was approved by the Advarra IRB (Institutional Review Board, human subjects committee, cirbi@advarra.com accessed on 23 December 2023) as a retrospective chart review of clinical records already available to the corresponding author prior to 1 March 2023. No additional testing was performed for the purpose of this study. The trio-WGS in our 50 subjects were all evaluated in the 15-month interval from December 2021 to February 2023, which is a few months following the last major update to Variantyx^®^ interpretation software (https://www.variantyx.com, accessed on 23 December 2023).

### 4.2. Sequencing and Data Analysis

Available clinical notes from all subjects were reviewed for phenotypic data. Any DNA variants identified in the official laboratory reports were tabulated for this study. In addition, raw genomic data from each subject were evaluated personally by the corresponding author. This included a comprehensive review of the raw data on the Variantyx^®^ bioinformatics platform accessible to laboratory personnel, including the Integrative Genomics Viewer (IGV, [47]) of any variants of interest to verify the presence of that variant and exclude artifacts. Any inherited variants of potential relationship to the patient’s phenotype, and all coding de novo variants, were recorded in the patients’ individual visit encounter notes. The DNA sequence data analysis pipeline is abstracted in Figure 1.

WGS analyses using Variantyx^®^ included genome-wide sequence analysis (for single-nucleotide variants, deletions/insertions, intronic, regulatory, and intergenic variants), genome-wide structural variant analysis (for copy number variants, duplications/deletions, regions of homozygosity, uniparental disomy, mobile element insertions, inversions, and aneuploidy), mitochondrial-genome sequence analysis (for heteroplasmy ≥5% and large deletion analysis), and short tandem repeat analysis (for the following genes: AFF2, AR, ATN1, ATXN1, ATXN2, ATXN3, ATXN7, ATXN8OS, ATXN10, C9ORF72, CACNA1A, CNBP, CSTB, DIP2B, DMPK, FMR1, FXN, HTT, JPH3, NOP56, NOTCH2NLC, PABPN1, PHOX2B, PPP2R2B, TBP, and TCF4). Additional information is available at https://www.variantyx.com (accessed on 23 December 2023) [48].

### 4.3. Gene Categorization

In the determination of diagnostic yield, we sought to be conservative in that each variant determined to be disease causal (PDVs) has a high probability of being so. Thus, we restricted PDV annotation to genes published with direct association with ASD. Genes were thus placed into two major categories: “Direct” and “non-Direct”. 

“Direct genes” are those with a direct published relationship to ASD, and are designed in Table 5 as A1 through A3 dependent on the estimated strength of that association, in particular using SFARI rankings [18] or AutDB evidence scores [19]. Overall, Direct genes are highly likely to be associated with ASD. “non-Direct genes” are those without a direct published relationship to ASD, and are designed as B1 through B3 (Table 5). Overall, non-Direct genes range from somewhat likely to unlikely to be associated with ASD (Table 5).

### 4.4. Variant Categorization

Variants were assigned as “Qualifying” if they are real (verified using IGV), coding (changing the amino acid code), rare (prevalence < 1/100), and evolutionarily conserved (at least moderate). Characteristics of different types of coding variants (e.g., missense, frameshift, deletion), and the importance of prevalence and conservation to variant annotation, can be found in a recent review (Tables 2–4 of [56]). Moderate conservation was assumed present if both PhyloP and PhastCons were >0.7 and assumed absent if both were <0.4. Otherwise, conservation was manually determined using the University of California Santa Cruz (UCSC) Genome Browser [57]. Splice-site variants were included if >0.6 on SpliceRF or SpliceADA. Thus, the focus of this study was on rare, high-penetrance variants. Common, low-penetrance variants require a much larger number of study subjects to evaluate, and thus were not considered. For instance, common variants of methylenetetrahydrofolate reductase (MTHFR) were excluded despite prior association [58]. Additionally, single-heterozygous variants in genes well categorized with autosomal recessive disease (e.g., potential carrier status) were not evaluated. 

In cases where one genetic variant is judged as sufficient to drive the bulk of disease causation, a PDV was assigned. PDVs were assigned as per the following five categories: De novo: Any very rare (<1/10,000), de novo, Qualifying variant in a Direct gene, with clinical correlation. Single-copy variants in genes with well-established autosomal recessive inheritance were excluded.X-linked: Any very rare (<1/10,000), inherited, hemizygous, Qualifying variant in a Direct gene on the X-chromosome, with clinical correlation.Autosomal recessive: Any rare (<1/100), inherited homozygous or in trans compound heterozygous Qualifying variants in a Direct gene on an autosome, with clinical correlation.Autosomal dominant: Any very rare (<1/10,000), inherited Qualifying variant in a Direct gene on an autosome, with clinical correlation, and with the parent harboring that variant being affected with significant neurodevelopmental disease. “Significant” was defined as substantially affecting their quality-of-life per the family and in the judgment of the corresponding author.Maternal inheritance: Any very rare (<1/10,000), Qualifying variant in a mitochondrial-encoded gene (mtDNA) with clinical correlation that is either heteroplasmic (with the minor allele present at 40–98%) and/or with a pedigree highly suggestive of maternal inheritance.

Characteristics of these modes of inheritance and their relevance to ASD can be found at (Table 1 of [20]). 

Clinical correlation indicates that the phenotype of the subject is a good match for the phenotype reported as associated with similar variants in that gene, as per general practice in Clinical Genetics. In the case that little to no clinical information is reported in the literature, clinical correlation is attempted based on other factors, such as the known mechanism of the protein’s function and tissue expression profiles. 

Of note, mtDNA has several differences from nuclear DNA, so some allowances needed to be made. For example, the determination of whether any mtDNA variant passes clinical correlation was difficult given the extreme protean findings associated with mtDNA. Therefore, we only counted cases that had mitochondrial-related clinical findings in four or more domains among neuromuscular, neurodevelopmental, neuropsychiatric, functional (e.g., pain, gastrointestinal, dysautonomic), endocrine, immunological, metabolic (laboratory signs of mitochondrial dysfunction), and enzymological (complex I or IV < 30% in muscle or buccal cells, the latter by MitoSwab^®^ (Religen^®^, Plymouth Meeting, PA, USA)). Maternal inheritance was determined using a Quantitative Pedigree Analysis (QPA, [33]). Evolutionary conservation in transfer-RNA genes was queried in both primary and secondary structures by http://trnadb.bioinf.uni-leipzig.de, accessed on 23 December 2023. Heteroplasmic variants < 20% were excluded as likely being of recent somatic origin. None of the mtDNA variants thus excluded were reported in MitoMap [59] as associated with disease. 

Qualifying gene variants in a nuclear Direct gene that did not meet PDV criteria were designated as CPMs. 

Statistical analyses were performed using a two-tailed Fisher Exact Test [60] and/or MedCalc^®^ Odds ratio calculator [61].

## 5. Conclusions

Our study reveals that high-confidence diagnoses can be assigned to the majority (68%) of individuals with WGS followed by comprehensive reanalysis of raw sequence data. De novo PDVs (disease causal) are common (50%) and constitute the bulk of diagnoses, most of which are previous unreported conditions. Indeed, 14 conditions are herein described for the first time among the 50 ASD subjects. Sequence reanalysis increased the yield of identifying a PDV from 28% to 68% of the subjects, or from 14% to 68% if “Uncertain” and “Likely Negative” laboratory reports are excluded. Thus, genetic laboratory reports, particularly those without clinically significant genetic variants, are insufficient for eliminating or identifying genetic causes of, and/or contributions to, ASD. Since most clinicians are solely reliant on the report generated by genetic laboratories, the implications are that many patients and their families are being insufficiently counseled and genetically investigated concerning important aspects of diagnoses and treatment of ASD. 

The methodology of this reanalysis is detailed in this report, and requires specialized expertise in both genomics and ASD. One option we found to be highly effective is close collaboration between a medical genomicist and a clinical specialist with knowledge and experience in treating ASD. Given the very high prevalence of ASD, this is going to require additional training/expertise among laboratory genomicists and clinical specialists. 

Genetic data obtained were actionable in terms of altering management in the majority of cases (37/50, 74%). This figure does not include less-tangible benefits such as ending the diagnostic journey, avoiding additional unnecessary testing, and genetic counseling for potential recurrence risks. In particular, treatment recommendations were provided in 33/50 subjects (66%), which is a function of the actionability of many of the relatively small number of common pathways involved in ASD, despite the large number of genes associated. These pathways lead to frequent recommendations for non-prescription treatments, generally mitochondrial-targeted (e.g., multiple nutrients and antioxidants) and/or cation-channel-targeted (e.g., potassium, magnesium) supplements, as well as prescription medications. Off-target diagnoses were few, and additional costs from testing and referrals were minimal, except for Immunology referral in 16%. Our data support the routine use of WGS with expert evaluation for cases with autism in general, as well as strengthening the scientific foundation of autism as potentially treatable in many cases.

Finally, our intriguing finding that de novo variants constitute the bulk of the identifiable genetic component to ASD, if validated in future studies, likely is important in the pathogenesis of ASD, and perhaps can lead to understanding of the rapid escalation in its prevalence.

## Figures and Tables

**Figure 1 ijms-25-01192-f001:**
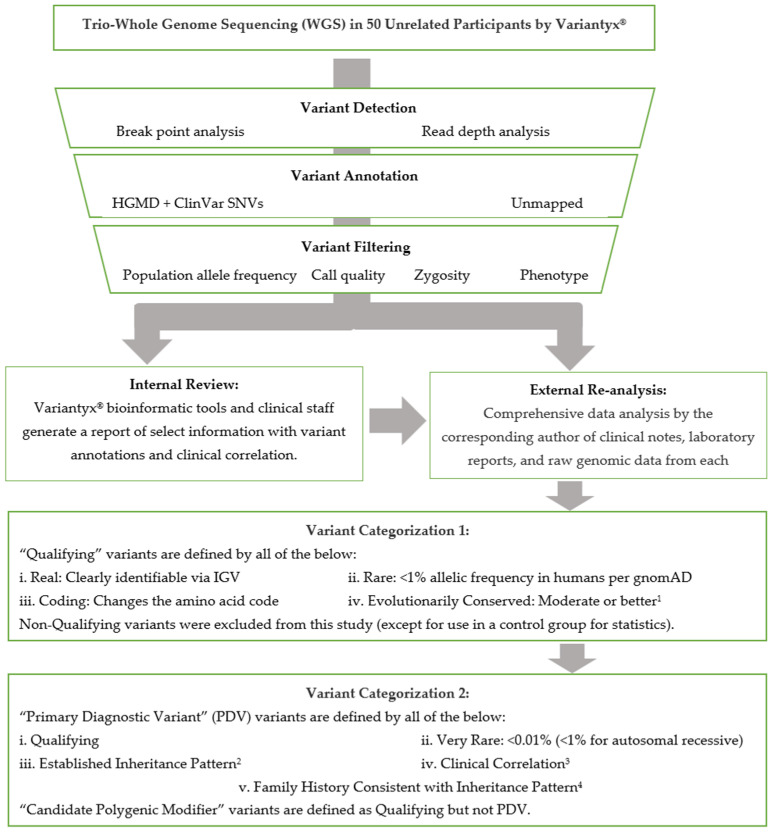
Variant Annotation Pipeline. ^1^ Moderate conservation, herein, corresponds to >90% amino acid concordance in mammals. See Methods section for details. ^2^ Monogenic only, including de novo, X-linked, autosomal recessive, autosomal dominant, maternal (mtDNA) inheritance. ^3^ Information available from subject is a good clinical match for phenotype described. This qualification was lowered for de novo variants with little to no reported phenotype (see text). ^4^ See Section 4 for details.

**Table 1 ijms-25-01192-t001:** Clinical manifestations in our subjects.

Subject #	Age and Sex	Developmental	Verbal ^1^	Seizure/EEG	Regression	Other Neuropsychiatric ^2^	Other Phenotypes ^3^
1	20M	severe ID	no	no/spike-wave	no, plateau		
2	10F	moderate ID	reduced	seizures	multiple	OCD, tics, anxiety	fatigue, pain, GI
3	14F	moderate ID	reduced	none	multiple	OCD, anxiety	GI
4	5M	moderate ID	reduced	none	no	anxiety	GI
5	22M	multiple LDs	yes	none	no	anxiety	
6	10M	severe ID (mild in sister)	no (yes in sister)	none (none)	one (one)		GI
7	13M	multiple LDs	yes	none	no		
8	6F	mild–moderate ID	reduced	none	one		
9	22M	severe ID	no	myoclonic	no		
10	5M	likely normal	very little	none	no		
11	12M	mild ID	yes	none	yes, chronic	choreiform, weakness, ataxia	systemic inflammatory response syndrome, immunodeficiency, autonomic instability, GI/parenteral nutrition, ventilation dependence
12	4F	severe ID	no	seizures	no		
13	26M	severe ID	NR	none	no	ataxia, choreoathetosis, cerebral palsy	
14	14M	multiple LDs	yes	none	episodic	OCD, tics	fevers to 40.5 C
15	19M	mild–moderate ID	NR	seizures	episodic	tics, psychosis/cyclical catatonia	
16	6M	multiple LDs	yes	none	one		
17	10M	severe ID	no	none	no		
18	12F	multiple LDs	yes	none	no	tics	
19	14M	moderate ID	no	none	multiple	tics	CVID
20	14M	normal	no	seizures	one	tics	
21	16M	moderate ID	yes	seizures	one	OCD, anxiety, depression	pain, cyclic vomiting
22	6M	severe ID	no	none	multiple		
23	5M	moderate ID	no	none	one	OCD, Brown syndrome	
24	25F	mild–moderate ID	yes	seizures	no	OCD, anxiety, psychosis	GI
25	16F	mild ID	yes	none	no	OCD, anxiety	fatigue
26	5M	moderate ID	yes	none	likely		fatigue, GI
27	16M	moderate ID	reduced	none	one	OCD, ADD	
28	22M	multiple LDs	yes	seizures	no	OCD, anxiety	
29	10M	moderate ID	yes	none	one	tics, ADHD	GI
30	13M	moderate–severe ID	yes	none	no		GI
31	3M	mild ID	yes	none	one		GI
32	12M	mild ID/LD	yes	none	no	OCD, ADHD, bipolar, POTS, PTSD	GI
33	7M	mild ID	yes	none	no	ADD, motor dyspraxia	GI
34	19M	severe ID	reduced	none	one	OCD, tics, ADHD	GI
35	22M	moderate ID	yes	seizures	no	ADHD	Noonan syndrome
36	8F	normal	yes	none	no	OCD, ADHD	fatigue
37	5M	mild–moderate ID	no	none	one		GI
38	17M	mild ID	reduced	none	likely	tics	hypogammaglobulinemia/CVID-like
39	12M	mild ID	yes	seizures	no	tics, ADHD	pain, GI
40	9M	mild LD	yes	none	no	OCD, tics, ADHD, anxiety	hearing loss, GI
41	10M	moderate ID/LD	yes	seizures	one	macrosomia	
42	13M	mild ID	reduced	neonatal only	multiple		fatigue
43	18F	moderate ID	yes	seizures	no	OCD, ADHD, bipolar, POTS, PTSD	fatigue, pain, GI
44	6M	severe ID	reduced	none	multiple	anxiety	
45	15M	mild–moderate ID	yes	seizures	no	ADHD	GI
46	16F	severe ID	no	seizures	multiple	OCD, anxiety, depression	Turner syndrome, GI, hypogammaglobulinemia
47	7M	severe ID	no	none	multiple	tics	
48	13F	mild LD	reduced	seizures	no	insomnia	GI, pain
49	7M	severe ID	yes	none	one	OCD, tics	GI
50	7F	likely normal	reduced	none	likely		

EEG = electroencephalogram, ID = intellectual disability, OCD = obsessive–compulsive disorder, GI = gastrointestinal3, LD = learning disability, NR = not recorded, CVID = common variable immunodeficiency, ADD/ADHD= attention deficit disorder without/with hyperactivity, POTS = postural orthostatic tachycardia syndrome, PTSD = post-traumatic stress disorder. ^1^ Reduced speech is part of the diagnostic criteria for autism; cases were flagged with light green background only when expressive speech was essentially absent. ^2^ Light blue highlighting in the penultimate column of Table 1 is for tics (13 cases, 26%) as a marker for potential PANS/PANDAS, as some level of obsessive traits is so common in autism that OCD is difficult to differentiate from background. Highlighting in other columns is explained in the text. ^3^ This is an incomplete listing limited to selected manifestations recorded in the clinical records available. Within, GI refers to gastrointestinal manifestations, which most often included reflux, bacterial overgrowth, and/or irritable bowel.

**Table 2 ijms-25-01192-t002:** Qualifying variants in our subjects.

Participant #	Family History	Inheritance Pattern of Primary Diagnostic Variants	Autism Genes with Qualifying De Novo Variants	Autism Genes with Qualifying Inherited Variants	Lab Identified ^1^
1	essentially negative	de novo	* **EHMT1** *	RYR2	yes
2	essentially negative	autosomal recessive, CompHet *in trans*		* IVD * *(AR-CompHet)*	yes
3	essentially negative	de novo	* PGAM5 *	CDH15	no
4	essentially negative	de novo	* COL4A1 *	TRAP1	no
5	essentially negative	de novo	* TRPM2 *		no
6	sister autism; siblings and mother ADHD	X-linked, mother carrier		*THOC2 (XL-Mat)*, dup with CHL1, RIC1 (AR-CompHet)	no
7	brother LD	de novo	* del 5 SFARI genes *		yes
8	essentially negative	de novo	* MTMR4 *	2p16.1p16.1 dup with FANCL, 9p13.3p13.3 dup	no
9	essentially negative	de novo	* **del with UBE3A** *	**FRMPD4** (XL)	yes
10	slow speech in twin sister, now normal				N/A
11	affected sister; parents with small fiber neuropathy	de novo	* USP20 *	CLPX, POLRMT, **RELN**, SCN10A	no
12	essentially negative	de novo	* SLC41A2 *	FLNA (XL)	no
13	brother LD, ADHD, possible autism	autosomal recessive, homozygous		* SLC1A4 * *(AR-Hom)*	yes
14	essentially negative			SCN9A, CPT2	N/A
15	essentially negative	de novo	* **KCNB1** *	MT-TW	yes
16	essentially negative	X-linked, mother carrier		*USP9X (XL-Mat)*, CIC	no
17	essentially negative	de novo	* GRIK1 *		no
18	brother LD; father possible autism			GABRA1, tetrasomy at 14q32q33 with 26 genes	N/A
19	essentially negative			**CACNA1A**, **RIMS1**	N/A
20	essentially negative				N/A
21	sister ADHD	de novo	* OCM *	SCN10A	no
22	sister ADHD; mother ADD	X-linked, mother carrier		*NLGN4X (XL-Mat)*, 4p16.1p16.1 dup with SORCS3, 9p24.3p24.3 dup with DOCK8, MT-ND3	yes
23	essentially negative			POLA1 (XL), **PAH** (AR-CompHet), 22q11.21 11.5-kb del, 14q32.33 27-kb dup, MT-CYB	N/A
24	essentially negative			7p22.3p22.3x1, **DYNC1H1**	N/A
25	essentially negative	de novo	* **ANKFN1** *	**SCN2A**, **ASH1L**	no
26	essentially negative	de novo	* CEP170 * *, NUP210*		no
27	essentially negative			SCN9A, RYR2	N/A
28	essentially negative	de novo	* **ANKRD11** *	TRAP1	yes
29	essentially negative				N/A
30	brother OCD, autism	de novo	* RIF1 * *, AGO3*	PRKCA	no
31	essentially negative				N/A
32	affected brother; father probable LD	de novo	* Large del with GGNBP2 *	SETD1B, **DEAF1**	yes
33	only extended relatives affected	autosomal recessive, homozygous		*EIF3F (AR-Hom)*, SCN10A	yes
34	father ADD	autosomal dominant, paternal		***TCF20** (AD-Pat)*, MT-TC	no
35	essentially negative	de novo	* KRAS *	MTHFR, HSPG2	yes
36	mother and dizygotic twin with autism	autosomal dominant, maternal		***RERE** (AD-Mat)*	yes
37	essentially negative			GBE1 (AR-CompHet)	N/A
38	essentially negative			MTHFR, **TSC2**, MT-CYB	N/A
39	father possible autism	de novo	* YTHDF1 *	CUX2, 148-kb del with IMMP2L	no
40	essentially negative	de novo	*GRB10*, *STAT1*	RYR2	no
41	essentially negative	de novo	* GOLGB1 *		no
42	essentially negative	de novo	* GABRA1 *	16q23.1q23.1, 736.50-kb with ADAMTS18	yes
43	essentially negative	de novo	* SP8 *	MT-CYB	no
44	mother and sister ADHD				N/A
45	essentially negative			SHROOM4 (XL)	N/A
46	brother ADHD			**KMT2E**, RYR2, MT-CO3	N/A
47	essentially negative			SCN4A, KCND2, 7q31.31q31.31x1 del with IGHG2	N/A
48	sister ADHD, ID, seizures, MELAS	de novo + mtDNA	* **SPEN** *	* MT-CO1 *	yes
49	brother autism	autosomal recessive, CompHet *in trans*	Large del with 53% PRODH	***ZNF292** (AR-ComHet)*, Xq22.3q22.3x1, 54 base pairs; includes TBC1D8B	no
50	essentially negative	de novo	* HSPA1A *		no

See Table 1 for some clinical abbreviations; CompHet = compound heterozygote, AR = autosomal recessive, XL = X-linked, Mat = maternally inherited, dup = duplication, del = deletion, Hom = homozygous, N/A = not applicable, kb = kilobase, AD = autosomal dominant, Pat = paternally inherited, MELAS = mitochondrial encephalopathy, lactic acidosis, and stroke-like episodes. Genes in *italic font* indicate PDVs. Dark red bold font refers to “A1” genes; Red font: “A2” genes; Orange font: “A3” genes; Green font: mtDNA genes. Unless otherwise noted, all variants are heterozygous on autosomes. ^1^ Lab identified indicates whether the variant was listed on the laboratory report.

**Table 3 ijms-25-01192-t003:** Primary Diagnostic Variants (PDV) that were and were not listed on the official laboratory report with protein function.

Participant #	Gene	Report ^1^	Disorder	NDD per HGMD ^2^	Protein Function	Cation Trans-port	Redox State	Amino Acids	Ubiquitination	Neurotransmitter	Gene Expression	Cell Division
On Lab Report												
1	EHMT1	Uncertain	Known		Histone methyltransferase						Yes	
2	IVD	Candidate POSITIVE	Known		Amino acid metabolism			Yes				
7	del SFARI x5 ^3^	POSITIVE	Known		Ubiquitination/cation channel/cholinergic receptor ^3^	Yes	Yes		Yes	Yes		
9	UBE3A ^3^	POSITIVE	Known		Ubiquitination		Yes		Yes			
13	SLCIA4	POSITIVE	Known		Amino acid transport			Yes				
15	KCNB1	POSITIVE	Known		Potassium transporter	Yes						
22	NLGN4X	Likely	Known		Neuronal cell-cell interactions, glutamate receptors					Yes		
28	ANKRD11	Likely Positive	Known		Transcription						Yes	
32	GGNBP2 ^3^	POSITIVE	Known		Growth suppressor?							
33	EIF3F	Uncertain	Known		Translation initiation factor						Yes	
35	KRAS	Likely Negative	Known		Ras protein, GTPase activity, regulation of cell proliferation							Yes
36	RERE	Likely Negative	Known		Transcription						Yes	
42	GABRA1	Likely Negative	Known		GABA receptor, chloride channel					Yes	Yes	
48	SPEN	Likely Positive	Known		Transcription							
48 (PDV2)	MT-CO1	Likely Negative	Known		Energy metabolism		Yes					
Not on Report												
3	PGAM5		Novel	1	Programmed cell death, mitophagy		Yes					
4	COL4A1		Known		Collagen, structural							
5	TRPM2		Very rare ^5^		Calcium channel, oxidative stress	Yes	Yes					
6	THOC2		Very rare ^6^		Transcription						Yes	
8	MTMR4		Novel		Ubiquitination, vesicular fusion, phagocytosis				Yes			
11 ^4^	USP20		Novel	2	Ubiquitination, inflammatory signaling				Yes			
12	SLC41A2		Novel	3	Cation 2+ transporter, including magnesium	Yes						
16	USP9X		Known		Ubiquitination, separating sister chromatids, axonal growth				Yes			Yes
17	GRIK1		Novel	3	Glutamate ionotropic receptor kainate type	Yes				Yes		
21	OCM		Novel	0	Calcium buffering	Yes						
25	ANKFN1		Nove l ^7^		Orientation of mitotic spindle and cell polarity							Yes
26	GEP170		Very rare ^8^		Centrosome component							Yes
30	RIFI		Novel	3	Cell check point							Yes
30 (PDV2)	AG03		Novel	3	Transcription						Yes	
34	TCF20		Known		Transcription						Yes	
39	YTHDFI		Novel		Binds m6A-containg mRNAs						Yes	
40	GRBIO		Novel	5	Involved in multiple cell signaling cascades							
41	GOLGB1		Novel	3	Golgi crosslinking							
43	SP8		Novel	0 ^9^	Transcription						Yes	
49	PRODH		Known	0 ^9^	Amino acid metabolism			Yes				
50	HSPA1A		Novel		Chaperone		Yes					

^1^ The text corresponds to the actual wording on the report in respect to that variant, and the shading reflects the color on the report. ^2^ HGMD = Human Gene Mutation Database (https://www.hgmd.cf.ac.uk/ac/index.php accessed on 23 December 2023). ^3^ Part of a contiguous gene deletion. Where a specific gene is listed, it is believed to be the Primary Diagnostic Variant (PDV). In subject 7, there are 5 different SFARI-listed genes in the deletion, and the PDV is unclear. ^4^ Another variant in this subject is discussed in Section 3.5. ^5^ One de novo was reported in ADHD, polymorphism which was associated with bipolar. ^6^ Reported in 4 unrelated families. ^7^ However, it was reported as part of contiguous genes’ deletions. ^8^ Reported 3 times, plus 3 more in contiguous gene deletions. ^9^ However, cases are reported with birth defects (subject 43) and autoinflammation (#49).

**Table 4 ijms-25-01192-t004:** Yield of genetic testing in autism spectrum disorder per prior reports.

Platform	Variant Types Detected	Subjects	Diagnostic Yield	De Novo Yield ^1^	Reference
FMR-1	FMR-1	50	2%		Shevell et al., 2001 [21]
FMR-1	FMR-1	502	1.3%		Munnich et al., 2019 [22]
Cytogenics	CNV, MECP-2	32	41%		Schafer and Lutz, 2006 [23]
Microarray	CNVs	29	27.5%		Jacquemont et al., 2006 [24]
Microarray	CNVs	1532	3.0%		Leppa et al., 2016 [25]
Microarray	CNVs	502	8.8%		Munnich et al., 2019 [22]
Targeted Sequencing ^2^	SNVs and indels	>11,730	5.7%		Stessman et al., 2017 [26]
Trio-WES	SNVs and indels	17	~90% ^3^	18%	Al-Mubarak et al., 2017 [29]
Trio-WES	SNVs and indels	80 simplex families	9.2% ASD, 6.7% suspected ASD	8%	Du et al., 2018 [27]
Trio-WES	SNVs, small indels, and CNVs	405	16%	15%	Miyake et al., 2023 [28]
Trio-WGS	Essentially all variants	32	50%	19%	Jiang et al., 2013 [30]
Trio-WGS	Essentially all variants	100	41%	14%	Abdi et al., 2023 [31]
Trio-WGS	Essentially all variants	101	33%	20%	Sheth et al., 2023 [32]

^1^ De novo yield indicates the percentage of subjects with at least one de novo variant thought by the authors to be likely to be disease related. Yields are only shown for large sequencing studies (WES/WGS). ^2^ Targeted sequencing of 208 candidate genes. ^3^ This study did not consider conservation or clinical correlation, and had a large yield of autosomal recessive findings consistent with the high rate of consanguinity in the Saudi population. FMR-1 = Fragile X mental retardation 1, CNVs = copy-number variants, SNVs = single-nulceotide variants, indels = insertions and deletions, trio = subject and parents, WES = whole-exome sequencing, WGS = whole-genome sequencing. This table is not comprehensive, particularly regarding the earlier studies/modalities, and is in part designed to abstract progress and yields over time.

**Table 5 ijms-25-01192-t005:** Gene Categories Indicating Known Association with ASD.

*Direct Genes: Direct Association with ASD:*
• **A1**—Indicating the highest association, it was designated to SFARI [18] 1 (or 1S “Syndromic”) ranking or a 4 -or 5-star AutDB [19] evidence score, whether ranked as such by those websites or by the present authors using their published criteria. • **A2**—Indicating genes with strong, but not overwhelming, association with ASD, it was designated to SFARI 2 (2S) ranking or with a 3-star AutDB evidence score, per those websites or the present authors using their criteria. • **A3**—Indicating the weakest level with direct association with ASD, it was designated to SFARI 3 (3S) ranking or with a 2-star AutDB evidence score as per those websites or the present authors. Additionally, some genes were placed in this category by the present authors due to findings of replicated or un-replicated statistical significance in association studies, as reported in ASD with one or more of the following: (i) an exonic de novo variant with >20 Combined Annotation Dependent Depletion (CADD) score [49,50] for genes related to another neurodevelopmental or neuropsychiatric disorder (such as bipolar disorder, schizophrenia, ADHD, and intellectual disability), (ii) a variant identified in a case with ASD [50] in a gene associated with another NDD, (iii) reported in ASD in ≥10 reported copy number variants (CNVs) per AutDB, and/or (iv) an ASD-like phenotype in an animal model. Lastly, some genes that qualified for the B1 category (as described below) became A3 genes if they were intolerant to loss-of-function mutations (supplemental material of [51], also seen in attachment 10 of [52]) and were either Fragile X syndrome genes that were found more enriched in an ASD group than a control group [53,54], also seen in attachment 4 of [52] or occurred in brain-expressed exons that were found with significant accumulation of de novo mutations in individuals with ASD when compared to controls [55] (also seen in attachment 1 of [52]).
*Non-Direct genes: Indirect or Absent Association to ASD:*
• **B1**—Indicating genes with an indirect association with ASD, was designated to (i) genes with a published direct association with any Direct gene, (ii) genes with direct association with another NDD phenotype that is itself associated with ASD (e.g., AD/HD, intellectual disability, schizophrenia, bipolar) with CADD ≥ 20, and/or (iii) genes in pathways in which ASD clearly has been associated. ASD-associated pathways include brain ion-channels, energy metabolism, amino acid metabolism, protein ubiquitination, neuronal cell development, cytoskeleton, epigenetic regulation, inflammation or immunodeficiency, and phosphatidylinositol signaling. • **B2**—Indicating genes with unknown association with ASD, designated to non-Direct genes neither meeting “B1” nor “B3” criteria. In practice, most “B2” genes occur in genes of uncertain function or in pathways with weak association with ASD. • **B3**—Indicating genes that are unlikely to be ASD related, it was designated to genes with known effects predominately in non-nervous tissues.

## Data Availability

The dataset is provided in Appendix A.

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
