# Peer review of "Reanalysis of Trio Whole-Genome Sequencing Data Doubles the Yield in Autism Spectrum Disorder: De Novo Variants Present in Half"

_ijms, 2024, doi:10.3390/ijms25021192_

Round 1
Reviewer 1 Report
Comments and Suggestions for Authors
Dear authors,
Thank you for sharing your data in the reanalysis of trio-WGS for autism to further elucidate the potential variants among 50 independent samples. The overall data was indeed novel and worthy for the standard of IJMS, and I have no reservations for publication with additional comments as below:
1) The authors stated the demographic data inclusive of severe and mild autism. How these variants are correlated clinically with the symptoms as well as in these two categories? is there any overlap between two variants and two classes of ASD, including the validated protein and affected neurological function?
2) Table 2. Qualifying Variants in Our Subjects, certain samples were not validated, such as samples #10,20,29, and 44. How do you discuss these samples?
3) Table 3 comprises good data showing the protein function of each of the Primary Diagnostic Variants (PDV). The data could be improved if it can be correlated with the clinical symptoms for each sample and further discussed in the discussion section.
4) To my reading, I think the authors made the discussion section as a re-iteration of the result section, so I would appreciate it if the authors could discuss the de novo variants in their data with the existing literature elsewhere. If there is a functional study to indicate these correlations, it would be for a discussion, to further support this study.
5) For section 4.1 (Subjects), did all 50 subjects pay for the service? Was a subsequent follow-up of genetic counselling conducted for the parents?
6) Please re-format the Table and font in the Table and figure to IJMS format. Thank you.
7) I would appreciate a figure to summarize the de nove variants revealed in this manuscript with the neurological pathways affected in ASD.
Author Response
Response to Reviewers, ijms-2816629, Reanalysis of Trio Whole Genome Sequencing Data Doubles the Yield in Autism Spectrum Disorder: De Novo Variants Present in Half
14-January-2024
The authors want to thank the reviewers for helpful comments which we believe make our paper significantly stronger.
Reviewer #1
Thank you for sharing your data in the reanalysis of trio-WGS for autism to further elucidate the potential variants among 50 independent samples. The overall data was indeed novel and worthy for the standard of IJMS, and I have no reservations for publication with additional comments as below:
1) The authors stated the demographic data inclusive of severe and mild autism. How these variants are correlated clinically with the symptoms as well as in these two categories? is there any overlap between two variants and two classes of ASD, including the validated protein and affected neurological function?
The authors consider genotype-phenotype correlations to be very important. We already did provide some of this data in the results section, of which we paste here:
2.5. Genotype-Phenotype Correlation
A PDV was identified significantly more often in subjects with neither tics nor OCD (22/26 (85%) versus the presence of either 12/24 (50%), P = 0.01, odds ratio 5.5, 95% CI 1.5-21). Neither manifestation alone was statistically significant, although there was a negative trend for PDV identification in subjects with tics (P = 0.08). All other clinical comparisons were non-significant, including no relationship with sex, the severity of ID, or the presence of seizures. A trend was observed for PDVs being less common in subjects with absent speech (6/13 (46%) v. 28/37 (76%), P = 0.08). In another potential trend, a PDV may be less common in patients with developmental regression (with 16/27 (59%) v. without 18/23 (78%), P = 0.2). Cases with and without a PDV are not significantly more or less likely to have an affected first-degree relative with a substantial NDD, although there is a possible trend (+PDV 13/34 (38%) with a first-degree relative affected, no PDV 3/16 (19%), P = 0.18, odds ratio 2.7, 95% CI 0.6-11).
However, the reviewers comment suggest that the above text was insufficient. Our molecular results could be considered to fall within three different categories: de novo, Mendelian inherited, and negative (no Primary Diagnostic Variant -PDV). The above analysis was performed by combining the first two categories together, which consist of the subjects in which a PDV was assigned. The reviewer appears to be suggesting that we lump the last two categories together, to compare those with and without a de novo mutation. Since there are actually quite few Mendelian cases, the results of these two approaches are essentially identical. In particular, there is no statistically significant or obvious trend regarding those with mild versus severe intellectual disability compared to those with or without a de novo variant (P = 1.0000). However, such correlations are difficult given the relatively small number of subjects in this study. Three sentences were added to the manuscript to reflect this (page 6).
2) Table 2. Qualifying Variants in Our Subjects, certain samples were not validated, such as samples #10,20,29, and 44. How do you discuss these samples?
The following text was added to the Discussion (page 16):
In four of the subjects (#10, 20, 29, and 44; 8% of the total) we did not find any variants that are likely correlated to autism by our methodology. There are many potential explanations, for this, including genes not yet associated with ASD, variants dismissed for one of various reasons as described in Methods that indeed are disease related, variants not well elucidated or evaluated by WGS using current practices (e.g., repeat areas, non-coding), multiple common variants, and/or epigenetic/environmental factors.
3) Table 3 comprises good data showing the protein function of each of the Primary Diagnostic Variants (PDV). The data could be improved if it can be correlated with the clinical symptoms for each sample and further discussed in the discussion section.
We agree, however, as explained in #1 above, there simply is insufficient power to do this, with 7 mechanisms times 3 main molecular result categories, in 50 subjects. In fact, another one hundred autism subjects (patients of Dr. Frye) are currently being evaluated by essentially the same methodology, and hopefully the numbers will be high enough to provide sufficient power for such correlations.
4) To my reading, I think the authors made the discussion section as a re-iteration of the result section, so I would appreciate it if the authors could discuss the de novo variants in their data with the existing literature elsewhere.
Excellent comment - another table, the new Table 4 (page 14), was added to summarize previous studies, and related text was added to the Discussion.
If there is a functional study to indicate these correlations, it would be for a discussion, to further support this study.
In terms of functional studies, none were performed. This is a retrospective chart review. Such studies would not be allowed by Ethics Committee guidelines. We have no grant or other funding that could support prospective investigation with functional studies (although we really would like to do that work!).
5) For section 4.1 (Subjects), did all 50 subjects pay for the service? Was a subsequent follow-up of genetic counselling conducted for the parents?
The following text was added to the manuscript:
The families’ insurance company paid for molecular testing in almost all cases as trio-WGS/WES in ASD is generally covered in the USA. In some cases, the insurance company paid for WES, and Variantyx provided WGS.
Results were discussed with the families at their next follow-up clinical visit, as is standard practice.
6) Please re-format the Table and font in the Table and figure to IJMS format. Thank you.
Done
7) I would appreciate a figure to summarize the de nove variants revealed in this manuscript with the neurological pathways affected in ASD.
This is very important, we looked, but 50 subjects simply have insufficient power for meaningful comparison. We intend to provide this information with the next study (already, 73/100 subjects have been reviewed; spoiler alert: results appear quite similar).
Reviewer 2 Report
Comments and Suggestions for Authors
n the present manuscript, it indicated that the reanalysis of trio whole genome sequencing (trio-WGS) using the Variantyx® bioinformatics platform revealed molecular diagnoses in Autism spectrum disorder (ASD). However, the manuscript is not well-written. I recommend that this paper accepted after minor revision.
The manuscript indicated that the trio-WGS detected de novo variants in contrast to the previous reports. However, it is difficult to understand the improved points because it is not well-described about the method of the previous laboratory reports. Also, it is not sufficient to describe about the method of trio-WGS such as variant detection including variant caller.
Author Response
Response to Reviewers, ijms-2816629, Reanalysis of Trio Whole Genome Sequencing Data Doubles the Yield in Autism Spectrum Disorder: De Novo Variants Present in Half
14-January-2024
The authors want to thank the reviewers for helpful comments which we believe make our paper significantly stronger.
Reviewer #2:
In the present manuscript, it indicated that the reanalysis of trio whole genome sequencing (trio-WGS) using the Variantyx® bioinformatics platform revealed molecular diagnoses in Autism spectrum disorder (ASD). However, the manuscript is not well-written. I recommend that this paper accepted after minor revision.
The manuscript indicated that the trio-WGS detected de novo variants in contrast to the previous reports. However, it is difficult to understand the improved points because it is not well-described about the method of the previous laboratory reports.
Excellent comment - another table, the new Table 4 (page 14), was added to summarize previous studies, and related text was added to the Discussion.
Also, it is not sufficient to describe about the method of trio-WGS such as variant detection including variant caller.
The authors had a conference call with the scientific people at Variantyx, the laboratory that did the sequencing. We were uncertain as to exactly what the reviewer was referring to, and we all felt that the methodology was described in detail, including in the Figure (page 21). In order to try to address the concern of the reviewer, an additional reference (new #48) was added, in which the Variantyx pipeline is explained in detail (referenced on page 20).